# Predominant Gram-Positive Etiology May Be Associated with a Lower Mortality Rate but with Higher Antibiotic Resistance in Spontaneous Bacterial Peritonitis: A 7-Year Study in a Tertiary Center in Romania

**DOI:** 10.3390/life15060855

**Published:** 2025-05-26

**Authors:** Sergiu Marian Cazacu, Ovidiu Mircea Zlatian, Elena Leocadia Plesea, Alexandru Ioan Vacariu, Mihai Cimpoeru, Ion Rogoveanu, Camelia Cristiana Bigea, Sevastita Iordache

**Affiliations:** 1Gastroenterology Department, University of Medicine and Pharmacy Craiova, Petru Rares Street no 2-4, 200349 Craiova, Romania; sergiu.cazacu@umfcv.ro (S.M.C.); ion.rogoveanu@umfcv.ro (I.R.); sevastita.iordache@umfcv.ro (S.I.); 2Microbiology Department, University of Medicine and Pharmacy Craiova, Petru Rares Street no 2-4, 200349 Craiova, Romania; ovidiu.zlatian@umfcv.ro; 3Doctoral School, University of Medicine and Pharmacy Craiova, Petru Rares Street no 2-4, 200349 Craiova, Romania; alexvacariu@yahoo.com; 4Emergency Clinic Hospital Craiova, 200642 Craiova, Romania; mcimpoeru810@gmail.com

**Keywords:** spontaneous bacterial peritonitis, antibiotic susceptibility, gram-positive bacteria, gram-negative bacteria

## Abstract

(1) Background: Spontaneous bacterial peritonitis (SBP) is associated with a 20% mortality and is mainly caused by Gram-negative bacteria (GNB); Gram-positive bacteria (GPB) were predominant in some areas; and increased antibiotic resistance was recorded. (2) Methods: A retrospective study was performed between 2018 and 2024. The type of isolated strains, antibiotic susceptibility, and mortality (in-hospital; 30-day; 90-day; and 1-year) were estimated; multivariate analyses evaluated predictive factors for in-hospital mortality risk. (3) Results: 45 culture-positive SBP, 28 culture-negative SBP, 6 bacterascites, and 670 control ascites were diagnosed; GPB represented 60%; two Candida peritonitis and 11 polymicrobial peritonitis (21.6%) were noted (without surgery; peritoneal dialysis; or tegumentary lesion). High resistance rates to cephalosporins and quinolones, and high carbapenem resistance for nosocomial GNB were recorded. A low resistance rate to Tigecycline was noted in all infection types; GPB was susceptible to Linezolid and Vancomycin; and GNB was susceptible to Aztreonam and Colistin. In-hospital mortality was 26.7% (40% for GNB-SBP; 20% for GPB-SBP), similar to culture-negative SBP (21.3%), and higher than in the control group (9%); long-term mortality remained higher. (4) Conclusions: microbial changes to GPB etiology and increasing resistance were noted, but with a lower mortality compared to GNB; higher mortality rates up to 1 year for culture-positive and culture-negative SBP were recorded

## 1. Introduction

The infections in liver cirrhosis are frequent and associated with significant short- and long-term mortality [1,2,3,4]; a trend of increased sepsis-related mortality in cirrhosis was noted [2]. Multiple mechanisms can contribute to the susceptibility to infection in cirrhotic patients: immune system dysfunctions, alterations in the enteric flora, bacterial translocation, and abnormalities of gut motility and the intestinal barrier as a result of portal hypertension [3,4,5]. The most frequent infections in liver cirrhosis are spontaneous bacterial peritonitis, pneumonia, urinary tract infection, and soft tissue infection [2,3,6]. A high prevalence of multidrug-resistant (MDR) infections was also noted in cirrhosis, which can lead to significant mortality [1,2,7].

Spontaneous bacterial peritonitis (SBP) represents the development of a bacterial infection of ascites in the absence of any intra-abdominal surgically treatable source of infection [8]; the prevalence is estimated at 1.5–3.5% in outpatients [8,9] and 10–30% in hospitalized cirrhosis patients [8,10]. In-hospital mortality is currently estimated at 20% [8], and 1-year mortality is estimated at 34% [8]. Three types of SBP were described: community-acquired (50%), diagnosed during the first 48 h after admission with no hospital admissions 90 days before the diagnosis, healthcare-associated (25%), diagnosed in the first 48 h but with a previous hospital admission during the last 90 days, and nosocomial SBP, diagnosed after 48 h of admission [2]. Some patients with cirrhosis can develop peritonitis due to the perforation or inflammation of an intra-abdominal (secondary bacterial peritonitis), which must be recognized because surgical treatment is needed [1]. Spontaneous fungal peritonitis (SFP) is rare but harbors a dismal prognosis [8,11,12], with 30% mortality [8], although 33–100% in-hospital mortality rates have been described [11,12].

The diagnosis of SBP/SFP is suspected by clinical symptoms and is confirmed by ascitic fluid examination. Clinical symptoms include abdominal pain and tenderness, vomiting, diarrhea, ileus, signs of systemic inflammation with fever, tachycardia or tachypnea, significant alteration of liver function or hepatic encephalopathy, shock, or renal failure [1,13]. Some SBP cases, especially in outpatients, may be asymptomatic. A neutrophil count above 250/mm^3^ in ascites is considered the gold standard for diagnosis [1,2,8]; cultures are positive in approximately 40% of cases [1], but bedside inoculation can increase the detection to 90% [2]. Cases with a PMN count above 250/mm^3^ and a negative culture are named SBP culture-negative and have a similar prognosis; culture-positive cases with less than 250 PMN/mm^3^ (bacterascitis) may represent a transient potential reversible colonization of ascites or a first step for the appearance of SBP [1,14].

The gut represents the main source of bacterial flora involved in SBP (*E. coli*, *Streptococcus* spp., *Staphylococcus*, *Enterococcus faecalis* and *faecium*, *Klebsiella* spp.). Most strains were Gram-negative bacteria (GNB) [1,2,15,16,17]. However, a microbial shift has occurred in the last two decades in many geographic areas, with Gram-positive bacteria (GPB) increasing and even surpassing Gram-negative species as the etiological agents of SBP, especially in Europe and America [11,17,18,19].

Our study aimed to assess the etiological spectrum of culture-positive ascitic fluid infection in our region, the rate of MDR bacteria, and the mortality rate in patients with liver cirrhosis and ascitic fluid infection. We hypothesized that GPB has become the predominant etiology in cirrhotic patients with ascitic fluid infection, and the antibiotic resistance also increases, which may change local guidelines in SBP therapy.

## 2. Materials and Methods

### 2.1. Clinical Data

We performed a retrospective study that included patients with culture-positive SBP admitted to the Emergency Clinical Hospital Craiova over 7 years (2018–2024). The study was approved by the Local Institutional Ethics Committee (protocol 10580/7 March 2025). All cirrhotic patients admitted to our hospital were analyzed, and patients with diagnostic paracentesis were initially selected. The diagnosis of ascitic fluid infection was confirmed by a positive culture obtained by inoculation on a dedicated culture medium. Patients with recent surgical intervention or peritoneal dialysis were excluded from the study. Cases with positive bacteriologic examination and neutrophil count above 250/mm^3^ were diagnosed as culture-positive SBP, whereas those with a positive culture and neutrophil count below 250/mm^3^ were designated as bacterascitis and included in the bacteriological analysis [1]; patients with negative cultures were diagnosed as culture-negative SBP (Figure 1). Because only one case of healthcare-associated SBP and one healthcare-associated bacterascitis were found, we divided SBP into non-nosocomial (paracentesis was performed less than 48 h after admission) and nosocomial (paracentesis after 48 h).

All cases with positive culture were analyzed for antimicrobial resistance, evaluated by antibiogram. An Excel database was created; for every isolated germ, the presence of MDR, XDR (extensive drug resistance), and the proportion of MRSA, ESBL, VRE, or Carbapenem-resistant bacteria were noted. The definition of MDR and XDR strains was based on the classification proposed by a joint group of experts from the European Center for Disease Prevention and Control and from Atlanta Centers for Disease Control and Prevention; MDR was defined as non-susceptibility to minimum one antibacterial agent from at least three antimicrobial categories, while XDR was defined as non-susceptibility to minimum one antibiotic in all but less than two antimicrobial categories [20]. For prognosis assessment, the main outcome analyzed was in-hospital and 1-year mortality, by using stratification risk factors for mortality such as age, etiology of SBP or SFP, clinical symptoms, comorbidities (hepatocellular carcinoma, portal vein thrombosis, cardiovascular and pulmonary diseases, diabetes, Clostridium difficile infection, COVID-19 infection, other infections, gastrointestinal bleeding), biochemical parameters (albumin, bilirubin, creatinine, Na, INR, hemogram), and the presence of renal failure and encephalopathy. Prognostic scores for liver failure, such as Child–Pugh–Turcotte (CTP) and creatine-CTP, MELD, MELD-3, and MELD-Na, were calculated and used for prognostic stratification.

### 2.2. Materials and Susceptibility Testing

The biological samples were analyzed according to the clinical laboratory microbiological protocol. All ascitic fluid samples were first examined microscopically, the number of polymorphonuclear cells was counted, and the presence or absence of Gram-positive or Gram-negative flora was recorded. The samples were immediately after reception inoculated in blood broth media, incubated for 18 h at 37 °C, and then inoculated on blood agar, chocolate agar, and MacConkey agar media, provided by BioMérieux (Salt Lake City, UT, USA). The culture media were again incubated for 18 h at 37 °C, and the plates were examined for bacterial growth. The bacterial strains were identified by culture properties, such as classical biochemical tests and antigen slide agglutination (e.g., *Staphylococcus aureus*). The antibiotic resistance testing was performed with a Kirby-Bauer disk diffusion test using antibiotic disks provided by Biomaxima S.A. (Cluj-Napoca, Romania). The ESBL and carbapenemase production of Enterobacteriaceae strains was detected phenotypically [21] using the disk diffusion double disk method with kits provided by Rosco Diagnostica (Albertslund, Denmark). Briefly, a 0.5 McFarland inoculum was deposited on a plate of Muller-Hinton agar, the five Rosco tablets (meropenem alone and in three inhibitor combinations plus temocillin), incubated overnight, then the zone-size differences were read according to manufacturer instructions to detect the presence of KPC, MBL, AmpC-porin, or OXA-48 carbapenemases.

### 2.3. Outcomes

The outcomes were represented by the etiology in patients with spontaneous bacterial peritonitis and antibiotic resistance for GPB and GNB. Although our primary microbiological analysis focused on culture-positive SBP, culture-negative SBP cases were included in the overall clinical and mortality analysis. We estimated the in-hospital and 1-year mortality rates for culture-positive, culture-negative, GPB-SBP, GNB-SBP, bacterascitis, and those with cirrhosis with uncomplicated ascites. This allowed a broader mortality assessment while maintaining microbiological precision for resistance analysis.

### 2.4. Statistical Analysis

We used Microsoft Excel (Microsoft Corp., Redmond, WA, USA) with the XLSTAT 2016 add-on for MS Excel (Addinsoft SARL, Paris, France) for data processing. For categorical variables, the percentages were calculated, while for continuous variables, because they did not follow normal distribution as assessed by the Shapiro–Wilk test, the median with interquartile range was estimated. We used the Mann–Whitney test for the assessment of the differences between groups for continuous variables, and the Chi-square test (or Fisher’s exact test when an expected cell count < 5) was used for categorical variables to assess the differences between groups regarding proportions.

The univariate and stepwise multivariate logistic analysis was performed to evaluate the factors associated with the risk and mortality in Gram-positive and Gram-negative SBP; factors identified through univariate logistic regression analysis with *p* < 0.30 were included in the multivariate model. Multicollinearity was evaluated using the variance inflation factor (VIF). We checked if all retained variables had VIF < 2, indicating no significant collinearity. Logistic regression assumptions were verified as follows: Linearity of logit (assessed via Box–Tidwell test), outliers/influential points (evaluated using Cook’s distance), and Goodness-of-fit (Hosmer–Lemeshow test; *p* > 0.05). To account for multiple testing, we applied the Benjamini–Hochberg FDR correction to univariate *p*-values. Only results with *p* less than the FDR threshold were significant after adjustment in the multivariate model. Kaplan–Meier curves were plotted for overall survival (culture-positive vs. culture-negative SBP controls) and bacterial classes (Gram-positive vs. Gram-negative SBP). Two-sided *p*-values < 0.05 were considered statistically significant.

## 3. Results

### 3.1. Characteristics of the Patient Group

Fifty-one patients with spontaneous bacterial/fungal peritonitis were recorded during 7 years; twenty-eight culture-negative SBP were also diagnosed during the same period from 749 patients with cirrhosis with ascites, and complete examination from diagnostic paracentesis. Forty-five patients had culture-positive SBP/SFB (two being Candida peritonitis), and six were bacterascitis; the mortality rate was 26.7% for culture-positive SBP/SFP, similar to 21.2% for culture-negative SBP (OR 1.35, 95%CI 0.47–3.92, *p* = 0.5799), and higher than in cirrhosis without infected ascites or SBP (9% in-hospital mortality, OR 3.67, 95%CI 1.79–7.47, *p* = 0.0003) or bacterascitis (0% in-hospital mortality, OR 4.85, 95%CI 0.25–92.58, *p* = 0.2939). Twenty-six infections were non-nosocomial (24 community-acquired and two healthcare-associated), and 25 were nosocomial (49%); because only two cases were healthcare-associated infections (one SBP and one bacterascitis), we grouped patients with ascitic fluid infection into nosocomial and non-nosocomial infections. The median age was 61.4 ± 11.5 years, 66% were males, and 76.5% had alcoholic cirrhosis. Most patients were Child B (43.1%) and C class (52.9%). Hepatic encephalopathy was diagnosed in 18%, hepatocellular carcinoma in 14%, portal vein thrombosis in 10%, and acute variceal bleeding in 12% of cases. The most frequently associated comorbidities were diabetes (10%) and Clostridium difficile colitis (6%). In-hospital mortality was 26.7%, being much higher in nosocomial (37.5%) than in non-nosocomial infection (14.3%, OR = 3.6, 95%CI 0.82–15.74, *p* = 0.0888); Thirty-day, 3-month, and one-year mortality were 44.4%, 46.1%, and 73% for SBP/SFP, and 0%, 16.7%, and 50% for bacterascitis (Table 1).

### 3.2. The Etiology of Ascitic Fluid Infection

We analyzed the etiology and susceptibility for all patients with ascitic fluid infection. Sixty bacterial strains and two fungal strains were responsible, and GPB were predominant (Table 2), thus confirming the bacterial etiological change. The temporal trend was relatively uniform, with GPB more frequent than GNB in almost all years (Figure 2).

### 3.3. Bacterial Antibiotic Resistance

We tested susceptibility to antibiotics for non-nosocomial- and nosocomial-associated ascitic fluid infection and Gram-positive and Gram-negative infection (Figure 3, Figure 4 and Figure 5, Appendix A).

For non-nosocomial infections, high susceptibility was noted for aztreonam, linezolid, carbapenems, piperacillin with tazobactam, and vancomycin (no resistance), followed by cefepime (9.1%), moxifloxacin and tigecycline (both with 13.3%), and cefotaxime (14.3%); no resistance was also noted for cefoperazone with sulbactam, ceftazidime with avibactam, nitrofurantoin, and quinupristin, although only a few strains were tested. Higher resistance was recorded for ampicillin (with or without sulbactam), ceftaroline, erythromycin, cefazolin, and oxacillin. For nosocomial infections, a low rate of resistance was noted for linezolid, aztreonam, colistin, and vancomycin (100% susceptibility), moxifloxacin (9.1%), chloramphenicol (12.5%), and tigecycline (21.3%); no resistance was also noted for two strains tested for ceftazidime with avibactam. Resistance rates of 50% or higher were noted for ceftriaxone, ciprofloxacin, imipenem, and trimethoprim with sulfamethoxazole (50%), clindamycin (53.8%), amikacin, ampicillin with sulbactam, cefuroxime, and tobramycin (57.1%), clarithromycin (58.3%), cefoxitin, ceftazidime, erythromycin, oxacillin, piperacillin with tazobactam, and meropenem (between 61–70%), cefotaxime and ceftaroline (75%), and cefazolin (83.3%). It is worth mentioning that, for some antibiotics, only specific bacteria were tested, such as vancomycin, clarithromycin, clindamycin, linezolid, and erythromycin for Gram-positive bacteria. In contrast, aztreonam, ampicillin + sulbactam, cefazolin, ceftazidime, cefoperazone + sulbactam, ceftazidime + avibactam, cefuroxime, and colistin were tested only for Gram-negative bacteria.

For Gram-positive bacteria, the antibiotics with the lowest resistance were linezolid and vancomycin (with no resistance from 35 and 13 strains tested, respectively), followed by moxifloxacin (11.5%), tigecycline (16.7%), chloramphenicol (18.5%), levofloxacin and ofloxacin (25%), doxycycline (26.9%), and ciprofloxacin (30.3%). Resistance rates of 50% and above were noted for amikacin (50%), erythromycin (56.7%), ceftaroline (60%), and oxacillin (64.5%). Lower rates for non-nosocomial compared with nosocomial infections were observed for levofloxacin (8.3 versus 41.7%) and tigecycline (10 versus 25%). For Gram-negative bacteria, the lowest resistance rates were observed for aztreonam (no resistance in both non-nosocomial and nosocomial infections), followed by colistin (7.1%, with only one non-nosocomial strain resistant), chloramphenicol (15.4%), and tigecycline (18.2%); the resistance in both non-nosocomial and nosocomial infections was similar for aztreonam, chloramphenicol, and tigecycline. Resistance rates above 50% for Gram-positive were noted for ceftazidime, ciprofloxacin, trimethoprim with sulfamethoxazole, and tobramycin. In the cephalosporin class, antibiotics with the highest in vitro susceptibility were ceftazidime with avibactam, cefoperazone with sulbactam, and cefepime (the last only in non-nosocomial infections).

For *Staphylococcus aureus*, no resistance was recorded for linezolid, low resistance was noted for moxifloxacin (13.6%), tigecycline (18.7%), doxycycline, chloramphenicol, and rifampicin (21.7, 22.7, and 26.1%, respectively), while resistance rates between 31 and 40% were found for ciprofloxacin, levofloxacin, gentamycin, trimethoprim with sulphamethoxazole, and tetracycline, 50% for amikacin, 54.2% for oxacillin, and between 58 and 64% for clarithromycin, clindamycin, ceftaroline, and erythromycin (Appendix A). Only one strain was tested for vancomycin, with no resistance noted. Significant differences between non-nosocomial and nosocomial infections were observed for amikacin, ciprofloxacin, and levofloxacin.

For *Streptococcus* spp., no in vitro resistance was noted for cefepime, cefotaxime, clarithromycin, clindamycin, chloramphenicol, moxifloxacin, linezolid, and vancomycin (Appendix A). Although the carbapenem resistance was tested in only two samples, no resistance was noted. The resistance to ceftriaxone, erythromycin, levofloxacin, ampicillin, tetracycline, and ofloxacin was between 25% and 50%, whereas resistance to cefoxitin, doxycycline, oxacillin, piperacillin, and trimethoprim-sulfamethoxazole was 100%. Non-nosocomial infection was associated with surprisingly more resistance to ofloxacin and tetracycline (66.7% versus no resistance for both antibiotics).

No in vitro resistance to ampicillin, quinolones, linezolid, gentamycin, tetracycline, tigecycline, or vancomycin was noted for non-nosocomial *Enterococcus* infection (Appendix A); all three strains were resistant to oxacillin, and one tested strain was resistant to teicoplanin. However, for nosocomial ascitic fluid infection, in vitro resistance appears in three of four strains tested for ampicillin, one of four strains tested for ciprofloxacin, two of four strains tested for gentamycin, and one of four strains tested for tetracycline. No resistance was noted for levofloxacin, linezolid, or vancomycin, regardless of the non-nosocomial or nosocomial infection type.

For *E. coli* infection, resistance was noted for ciprofloxacin (16.7%), gentamycin (20%), cefazolin (25%), trimethoprim with sulfamethoxazole (40%), and ampicillin (100%). No resistance was present for amikacin, aztreonam, ampicillin with clavulanic acid or with sulbactam, piperacillin with tazobactam, cephalosporins (except for cefazolin), chloramphenicol, colistin, levofloxacin, carbapenems, tetracycline, tigecycline, and tobramycin (Appendix A). 

In *Klebsiella* infection, no resistance (regardless of the non-nosocomial or nosocomial type) was noted for aztreonam, cefoperazone with sulbactam, ceftazidime with avibactam, colistin, chloramphenicol, and tigecycline (Appendix A). A resistance between 25 and 50% was reported for amikacin, cefepime, gentamycin, carbapenems, trimethoprim with sulfamethoxazole, and tetracycline, but all resistant strains were nosocomial (except one non-nosocomial strain resistant to tetracycline). A resistance between 60 and 100% was noted for ampicillin with clavulanic acid or with sulbactam, most cephalosporins (except for cefepime), ciprofloxacin, tobramycin, and piperacillin with tazobactam.

For *Acinetobacter baumanii*, antibiotics with the highest susceptibility were doxycycline, tigecycline, and colistin, with no in vitro resistance. The resistance to ampicillin, tetracycline, and cephalosporins (including 4th-generation cefepime) was 100%, to amikacin, gentamycin, tobramycin, and imipenem was 66.7%, and to trimethoprim + sulfamethoxazole was 50% (Appendix A).

A summary of antibiotic resistance for the main isolated bacteria is presented in Table 3.

We also evaluated the presence of multidrug-resistant (MDR) and extensively drug-resistant (XDR) bacteria, ESBL (Enterobacteriaceae species with beta-lactamase), *Staphylococcus aureus* MRSA, and VRE. MDR strains were present in 36 of 60 bacteria (60%), and XDR strains in three cases (5.1%), with two *Acinetobacter* spp. and one *Enterobacter* spp. being XDR strains. MRSA was noted in 12 cases (48%) and MSSA in 13 cases. Gram-negative ESBL strains were found in four of 21 bacteria (19%); carbapenem-resistant *Enterobacteriaceae* were present in four of 21 Gram-negative strains (19%) and two of four ESBL strains (50%). No vancomycin-resistant Enterococci (VRE) were found.

The assessment of overall resistance using the multiple antibiotic resistance index in Gram-negative bacteria showed that *Acinetobacter* spp. and *Klebsiella* spp. had the highest resistance, followed by *Pseudomonas* spp., which also had a great variability, as some strains showed almost no resistance and others had very high resistance. *E. coli* presented the highest antibiotic susceptibility (Figure 6).

### 3.4. Risk Factors for Gram-Positive Spontaneous Peritonitis and Mortality

We performed a univariate and multivariate analysis to evaluate the risk factors and mortality in Gram-positive and Gram-negative SBP.

For Gram-positive infection risk, univariate analysis has found no association with age (*p* = 0.740), gender (*p* = 0.673), previous antibiotherapy (*p* = 0.870), previous paracentesis (*p* = 0.931), previous PPI use (0.505), encephalopathy (*p* = 0.282), AKI (*p* = 0.608), HCC (*p* = 0.747), portal vein thrombosis (*p* = 0.762), gastrointestinal bleeding (*p* = 0.414), diabetes (*p* = 0.902), COVID-19 infection (*p* = 0.320), *Clostridium difficile* infection (*p* = 0.270), and non-nosocomial or nosocomial type (*p* = 0.412). Alcoholic etiology seems more frequent in GPB infections, although statistical significance was not attained (OR 4.18, 95%CI 0.66–26.42, *p* = 0.109). The presence of arterial hypotension was less likely in patients with GPB infection (OR 0.11, 95%CI 0.11–0.98, *p* = 0.036), and the mean pulse was also lower (but without statistical significance). Similar mean laboratory values were also found. In multivariate analysis, factors with *p* < 0.30 were selected; however, only the absence of hypotension was a predictive factor for GPB spontaneous peritonitis (Table 4).

To assess survival, the Kaplan–Meier curve was constructed (Figure 7 and Figure 8), and univariate followed by multivariate analysis was performed; risk factors included in univariate analysis were age group, gender, previous antibiotic and PPI usage, prior paracentesis, hypotension at admission, Child class C versus A/B, acute kidney injury (AKI), history or concomitant hepatocellular carcinoma, cardiovascular disease, portal vein thrombosis or diabetes, upper gastrointestinal bleeding during admission, *Cl. difficile* infection, nosocomial or non-nosocomial infection, and Gram-positive versus Gram-negative bacteria (Table 5). Male gender, hypotension at admission, AKI, HCC, *Cl. difficile* infection, nosocomial or non-nosocomial type, and Gram-negative PBS had a *p*-value lower than 0.25 and were included in the multivariate analysis; only male gender (OR 15.90), AKI (8.45), and Gram-negative infection (OR 7.71) were associated with in-hospital mortality, while for nosocomial infection, statistical significance was not attained, possibly because of the small number of patients (OR 10.83, *p* = 0.091).

We constructed Kaplan–Meier curves for culture-positive versus culture-negative SBP versus control, and for Gram-positive versus Gram-negative bacteria (Figure 7 and Figure 8). Mean survival was 101 days for culture-positive SBP (95% CI 41.3–160.4 days), 311 days for culture-negative SBP (120.5–501.9), and 550 days for the control group (502.6–599.2), with statistical significance (log-rank *p* < 0.0001). For Gram-positive SBP, the mean survival was 158 days (62.8–254.1), while for Gram-negative SBP, the mean survival was 39 days (0–94), also with statistical significance (log-rank *p* = 0.037). These findings suggest that the long-term survival for culture-positive SBP was lower than for culture-negative SBP and the control group, and also that the long-term survival for Gram-negative SBP was lower than for Gram-positive SBP.

## 4. Discussion

Cirrhotic patients are vulnerable to developing various bacterial, fungal, or parasitic infections because of a low immune status, dysbiosis, and intestinal barrier dysregulation [3,4,5,14,22]. SBP represents one of the most frequent infections in liver cirrhosis and is associated with a worse prognosis; in the last decade, new provocations represented by the etiological microbial shift and antibiotic resistance have appeared. Studies have shown an increase in MDR bacteria in both liver cirrhosis [7], and SBP [23], and a progressive rise in Gram-positive was also noted [24,25,26,27,28,29]; the shift may be uneven in various geographic areas, as a result of antibiotherapy use and abuse, instrumentation, and other factors [7,18,30]. The increased prevalence of MDR and XDR species has imposed an adaptation of the current guidelines regarding empirical treatment for SBP until antibiogram results are available [1,2]. Using a less efficient antibiotic because of less-known local susceptibility can be associated with higher mortality [2].

In our study, the percentage of culture-positive SBP/SFP was 61.6%, and the rate of SBP/SFP from total paracentesis was 9.7%, both in line with the literature data [1,2,14,31]. The rate of bacterascitis (or non-neutropenic ascitic fluid infection) from total infection was 11.6%, lower than in the literature data [26,32]; a meta-analysis of 16 studies has found an equal proportion of SBP and bacterascitis (both with 2%) [32]. Gram-positive bacteria represented more than 60% of the causes (65.6% in non-nosocomial and 60% in nosocomial infections), surpassing Gram-negative infection, similar to other studies when Gram-positive bacteria were predominant [24,26,27] or almost equal with Gram-negative flora [25,33,34,35]; we did not find any risk factors associated with this shift, because no instrumental maneuver was noted in patients’ recorded files, no previous paracentesis was recorded, and discharge data showed no skin lesion which could explain any contamination. It is worth mentioning that the bacterial shift toward Gram-positive etiology is an ongoing and unevenly spread worldwide phenomenon [18], with studies from Australia [26], China [35], France [24,36], Germany [34], Greece [27], Portugal [25], and USA [33] recording a predominance or equal proportion of Gram-positives, whereas other studies from Brazil [7], China [30,37,38], Romania [39], and Saudi Arabia [40] have still noted Gram-negative bacteria’s predominance.

Our study has also noted a high percentage of polymicrobial peritonitis (21.6%); all patients had no surgery performed, no skin lesions, and the flora was represented by both Gram-positive and negative (59.1 and 40.9%, respectively), with the most frequent bacteria represented by MRSA *Staphylococcus aureus* (six cases), followed by *Enterococcus* spp. (four cases), and *Klebsiella* and *Streptococcus* (three cases each); no contamination flora, such as coagulase-negative *Staphylococcus*, was identified. Two patients had *Clostridium difficile* infection during hospitalization, one patient had rectal adenocarcinoma with radio-chemotherapy, another had acute variceal bleeding, another had diabetes, and six patients had no other comorbidities except liver disease. The proportion of polymicrobial SBP in different studies was between 1.6 and 15.7% [25,26,30,33,35,36,40]. The association between *Clostridium difficile* infection and polymicrobial peritonitis in our study was interesting and may be related to the increased intestinal permeability in the case of pseudomembranous colitis.

Antibiotic resistance has increased during the last decades [1,2], especially in healthcare settings, and the impact of empirical antibiotic therapy was also significant. In previous guidelines, third-generation cephalosporins (especially Cefotaxime) were the preferred first-line therapy in SBP; the appearance of MDR (especially in nosocomial or healthcare-associated SBP) and the late occurrence of a microbial shift to Gram-positive can represent a significant challenge to the empirical treatment of SBP (and especially in culture-negative SBP, where a second paracentesis is very important to evaluate the therapeutic response). For this reason, an accurate evaluation of microbial susceptibility to antibiotics is significant because both etiological changes and corresponding susceptibility in SBP have geographical particularities. In our study, the resistance to the most-used cephalosporins was 9–33.3% in non-nosocomial infections (with values up to 50% for ceftaroline) but was 41–83.3% for nosocomial infections; a significant limitation was related to the fact that tests were mostly confined to Gram-negative species. The 4th generation cephalosporin cefepime was active in all Gram-positive infections (nosocomial and non-nosocomial) and also in non-nosocomial Gram-negative infections (with resistance rates between 0 and 12.5%); however, the testing in Gram-positive infections was limited. Several studies have shown increased resistance to cephalosporins and a potential limitation of indications to the community-acquired SBP and areas of low antibiotic resistance [41,42,43]. The resistance to quinolones was also significant, maybe related to the extensive use in urinary tract infections [44], with values between 23.5 and 66.7% for ciprofloxacin, and between 8.3 and 42.9% for levofloxacin (with lower than 25% resistance only for non-nosocomial GPB); one exception was moxifloxacin, with resistance rates between 9.1 and 13.3%, but it was tested only for Gram-positive bacteria. Regarding the aminoglycoside class, both amikacin and gentamycin were associated with significant resistance in both Gram-positive and Gram-negative infections, and also in non-nosocomial and nosocomial forms, with resistance rates as low as 20% for amikacin in Gram-positive non-nosocomial infection and as high as 75% in Gram-positive non-nosocomial infection. Tigecycline was tested in Gram-positive and Gram-negative (non-nosocomial and nosocomial) infections, with low resistance rates (10–25%) and low variations between non-nosocomial and nosocomial settings, which may suggest an important role in treating all SBP regardless of the type. Linezolid and vancomycin were associated with the highest susceptibility in Gram-positive infections (with no resistance noted); the same was recorded for aztreonam and colistin in Gram-negative infections (no resistance was detected for aztreonam, and resistance rates were between 0 and 16.7% for colistin). In the carbapenem class, no resistance was detected in Gram-positive (non-nosocomial and nosocomial) and Gram-negative non-nosocomial infections, but 50–70% resistance was found in Gram-negative nosocomial infections; the appearance of carbapenem-resistant *Enterobacteriaceae* was associated in the literature with a higher mortality rate [45]. The combinations of ampicillin with clavulanic acid or with sulbactam, and of piperacillin with tazobactam were less analyzed in our data; resistance rates between 42.9 and 66.7 were found, especially for Gram-positive and nosocomial infections, which can make these combinations less suitable for empirical treatment in our region. The literature suggested a combination of piperacillin-tazobactam-linezolid as an efficient alternative to piperacillin-tazobactam therapy [33]. Another interesting finding may be related to chloramphenicol (a less-used antibiotic), with significant testing, resistance rates between 10 and 23.5% in all settings, and less variability; however, potential side effects, especially in advanced cirrhosis, may significantly limit the usage in empirical therapy.

For *Staphylococcus aureus*, antibiotics associated with the highest in vitro susceptibility were linezolid, moxifloxacin, and tigecycline (vancomycin was not tested); the resistance to oxacillin was 53.8% in non-nosocomial and 54.5% in nosocomial infections. *Streptococcus* spp. was susceptible to many antibiotics (cefepime, cefotaxime, clindamycin, chloramphenicol, linezolid, vancomycin, and carbapenems). *Enterococcus* spp. was susceptible to quinolones (levofloxacin and partially ciprofloxacin), linezolid, tetracycline, and vancomycin. *E. coli* remains susceptible to many antibiotics (most cephalosporins, quinolones, colistin, carbapenems, tetracycline, and tigecycline), while for *Klebsiella* spp., significant resistance to cephalosporins, carbapenems, piperacillin, and tobramycin was noted for nosocomial infections. In *Acinetobacter baumanii* infection, high resistance rates were recorded for cephalosporins, quinolones, and tobramycin (regardless of nosocomial or non-nosocomial setting), and also for carbapenems and gentamycin in nosocomial infections; the in vitro susceptibility for colistin, doxycycline, and tigecycline remains high.

The presence of ESBL-producing strains was 19% in our study, similar to the literature data; a study has found a 49.4% rate of ESBL-producing strains for *E. coli* and 15% for *K. pneumoniae* [38]; the presence of ESBL was associated with a higher mortality rate [26,46]. The presence of carbapenem-resistant *Enterobacteriaceae* (CRE) was recorded in 19% of all GNB and 50% of ESBL strains; in a study, CRE was noted only in two *Enterobacteriaceae* infections, but 25% of *Enterobacter faecium* strains were vancomycin-resistant [26]. MDR rate was significantly higher (60%), but the proportion of XDR strains was lower (5.1%) than in a study of 70 SBP and 60 bacteriemia ascites, in which 20.8% of bacteria were MDR, and 10% were XDR [27]. No vancomycin-resistant *Enterococci* (VRE) were found; the appearance of VRE species in cirrhosis has been noted [27], although with a low prevalence [14] and an uncertain role in mortality [47].

The presence of SFP in our study was 3.9%, both being *Candida albicans* infections; the prevalence was in line with the literature data [8,11,12,26,48,49]. Both patients had alcoholic cirrhosis with multiple hospital admissions before *Candida* infection, but no ICU admissions were recorded before the moment of infection. One patient was admitted for upper gastrointestinal bleeding and had Child C cirrhosis (14 points), while the other had a previous urinary infection treated with third-generation cephalosporin. Both patients were treated with fluconazole; the evolution was favorable with no in-hospital mortality in both cases, and one patient survived more than one year after discharge.

In-hospital mortality rate for SBP was 26.7%, similar to the literature data [1,2,11,14,50,51]; it was 40% for Gram-negative SBP, 20% for Gram-positive SBP, 33.3% for Mixed-SBP, and 0% for SFP. A large, population-based US study has found 17.6% in-hospital mortality for SBP [50]. In our study, the mortality seems similar for culture-positive and culture-negative SBP in both short-term (in-hospital and 30-day) and long-term settings (90-day and 1-year), with higher mortality in both short- and long-term settings compared to control ascites (no SBP and no infection). The 1-year mortality rates were 73% for culture-positive SBP and 73.9% for culture-negative SBP versus 41% for the control group (OR 3.88, 95%CI 1.83–8.24, *p* = 0.0004 for culture-positive SBP, and OR 4.08, 95%CI 1.57–10.56, *p* = 0.0038 for culture-negative SBP). Some literature data have shown, however, a similar long-term prognosis for patients with ascites with and without SBP; in a study in Taiwan, the 30-day mortality rates for SBP and ascites without SBP were 24.2% and 14.1%, and 3-year mortality rates were 66.5%, and 61.1% [52]; another Danish study has found similar mortality rates after 4 months of admission for SBP and ascites without SBP [53]. The mortality is higher in nosocomial than in non-nosocomial SBP [54].

In Romania, several studies have assessed etiology in patients with SBP. A study in Constanta county has found 42 culture-positive and 30 culture-negative SBP over 10 years (2010–2019), with Gram-positive bacteria predominant (most frequent being *E. coli*), but only therapeutic data were presented (most frequent antibiotic used being cephalosporins), and no susceptibility information was available [39]. One study in Iasi has found eight Gram-negative and four Gram-positive bacteria, with four of eight Gram-negative bacteria resistant to ceftazidime (all susceptible to carbapenems) [55].

Our multivariate analysis failed to reveal any risk factor associated with Gram-positive versus Gram-negative SBP. In-hospital mortality was associated in multivariate analysis with Gram-negative SBP, male gender, and acute kidney injury. The nosocomial form of SBP was also associated with a higher risk of in-hospital death, although the statistical significance was not attained (OR 10.83, *p* = 0.091); the small sample size may be a significant factor. Male gender was strongly associated in our study with in-hospital mortality in SBP; the net prevalence of alcoholic cirrhosis in males, with associated immunosuppression, can be an explanation. A large study in the US has found, however, a small risk of mortality in women (OR 1.08, *p* = 0.03) [50]. Multivariate analysis in our study may be, however, significantly impacted by the small sample size. In the literature, acute kidney injury, nosocomial infection, sepsis, and encephalopathy were associated with death risk in SBP [25,38,50,51]; a meta-analysis of seven studies reporting the outcome of nosocomial versus non-nosocomial SBP has shown a pooled odds ratio of 2.78 for mortality in nosocomial SBP [56]. In some studies, *Acinetobacter*, *E. coli*, *Enterococcus*, and *Klebsiella*-associated SBP were associated with higher mortality [37,57], while *Streptococcus* spp. infection was associated with a lower mortality rate [57,58].

This study has several limitations. First, its unicentric and retrospective design resulted in a relatively small cohort, which restricts the generalizability of the findings. Although a multicentric approach or longer study duration could increase sample size and statistical power, such extensions are challenged by potential regional differences in microbial etiology and resistance profiles, as well as dynamic shifts in causative pathogens. Second, the possibility of contamination could not be completely excluded, although only one case of coagulase-negative Staphylococcus was identified, suggesting minimal impact. Third, the exclusion of culture-negative SBP from the primary microbiological analysis may have introduced selection bias, as these cases represent a substantial proportion of SBP presentations. However, culture-negative SBP was included in the clinical and mortality analyses to mitigate this bias. Fourth, the relatively small sample size may limit the robustness of the multivariate analyses, particularly for long-term mortality outcomes, and may amplify type II error risks. Fifth, a significant methodological limitation is the lack of genomic analyses, such as whole-genome sequencing (WGS), which would have allowed for a more comprehensive characterization of antimicrobial resistance (AMR) mechanisms and pathogen-relatedness. The absence of such genomic data restricts the depth of understanding regarding resistance gene carriage, clonal spread, and epidemiological linkages, particularly in multidrug-resistant and nosocomial isolates. Finally, although integration with broader institutional resistance surveillance is recommended, our findings highlight the need for continuous regional monitoring to inform antibiotic stewardship and therapeutic strategies.

## 5. Conclusions

A microbial change to a predominance of Gram-positive bacteria has been noted in our study, for both nosocomial and non-nosocomial settings; in nosocomial SBP, a high resistance of isolated strains to commonly used antibiotics such as most cephalosporins, quinolones, and aminoglycosides was also detected, and resistance to carbapenems was noted for Gram-negative nosocomial infections. Tigecycline had low in vitro resistance rates in both Gram-positive and Gram-negative (non-nosocomial and nosocomial) infections. In Gram-positive infections, linezolid and vancomycin were associated with the highest in vitro susceptibility, whereas in Gram-negative infections, aztreonam and colistin had the highest in vitro susceptibility (although the nephrotoxic potential of colistin renders its use in SBP as a last resort therapy and only in complex, resistant cases without renal dysfunction). In-hospital mortality rate was 26.7% for SBP (40% for Gram-negative SBP, 20% for Gram-positive SBP), similar to culture-negative SBP (21.3%), and higher than in bacterascites and the control group; long-term mortality remains higher than in the control group for both culture-positive and culture-negative SBP. In our study, Gram-negative etiology in SBP, acute kidney injury, and male gender were independent factors for mortality, while the nosocomial type of SBP, although it had a very high OR (10.83), failed to achieve statistical significance, possibly because of the small number of patients. Retrospective design and small sample size in our study may overstate the role of Gram-negative versus Gram-positive strain type in mortality risk; larger or multicenter studies are warranted to clarify the role of the bacterial strain in prognosis assessment.

## Figures and Tables

**Figure 1 life-15-00855-f001:**
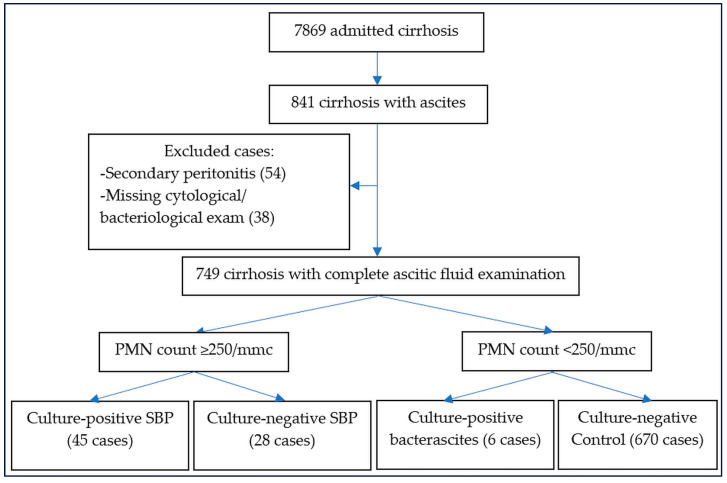
The flowchart for patients enrolled in our study.

**Figure 2 life-15-00855-f002:**
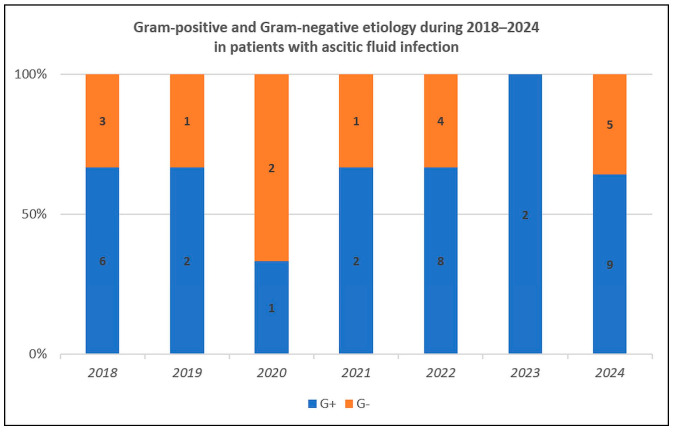
Temporal distribution for Gram-positive and Gram-negative strains during 2018–2024.

**Figure 3 life-15-00855-f003:**
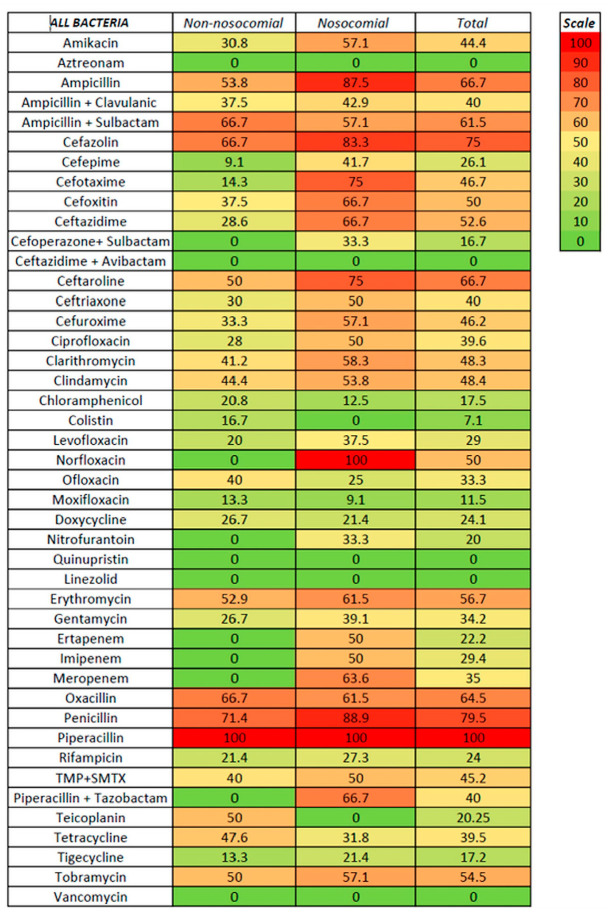
Antibiotic resistance in nosocomial and non-nosocomial ascitic fluid infection.

**Figure 4 life-15-00855-f004:**
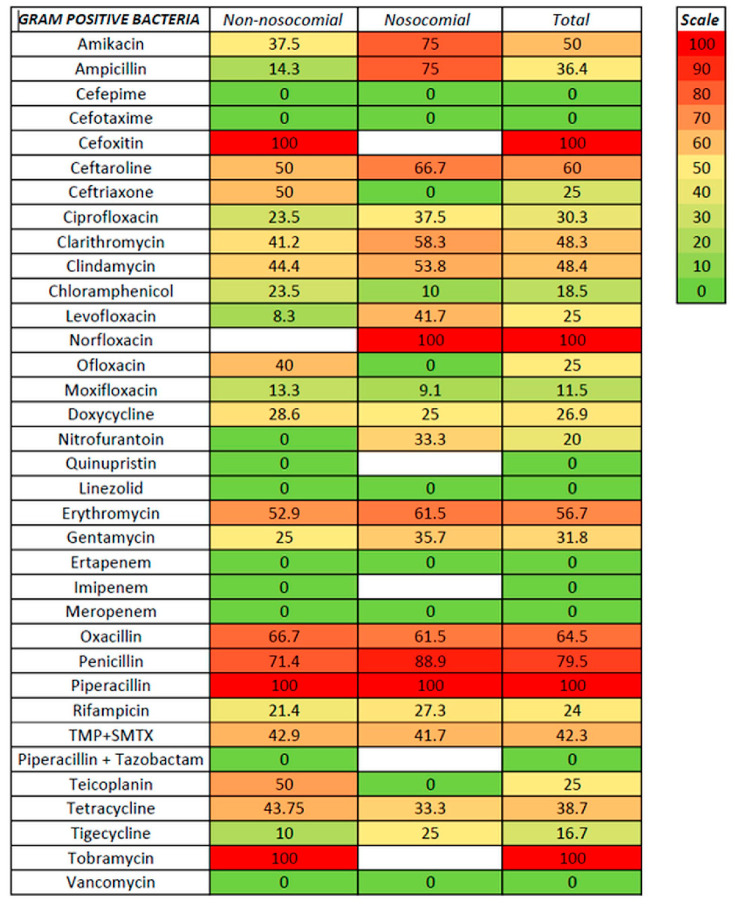
Antibiotic resistance in GPB ascitic fluid infection.

**Figure 5 life-15-00855-f005:**
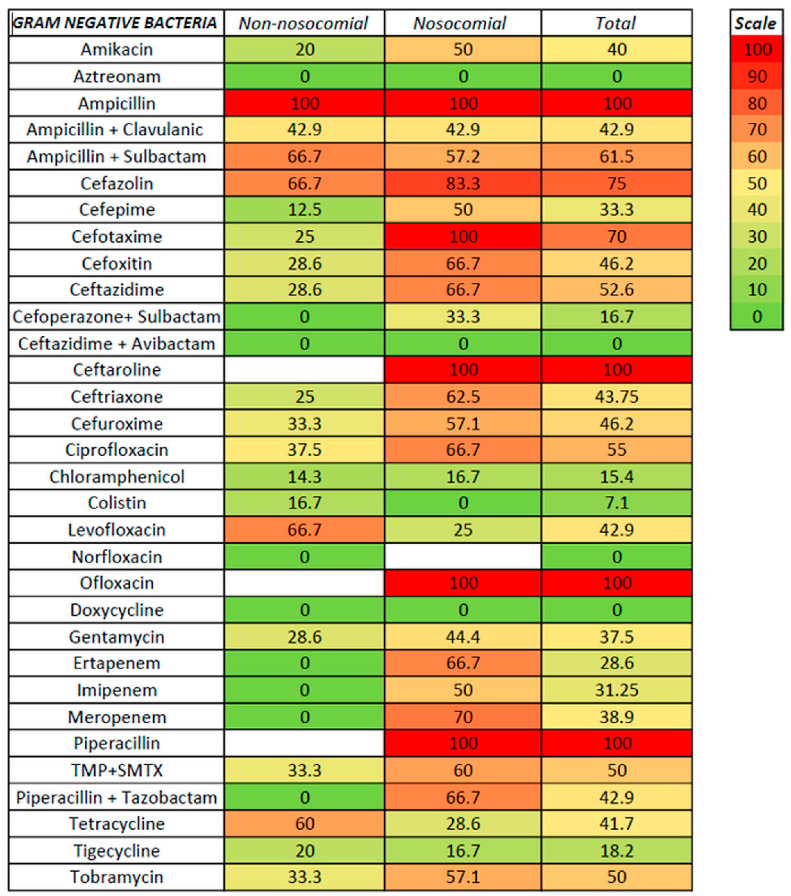
Antibiotic resistance in GNB ascitic fluid infection.

**Figure 6 life-15-00855-f006:**
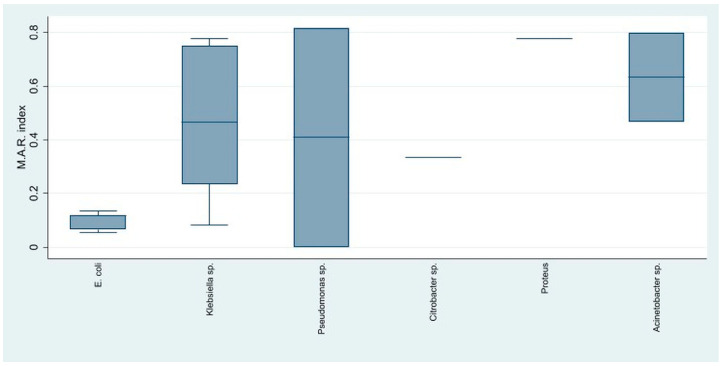
The distribution of the antibiotic resistance index among the Gram-negative identified strains.

**Figure 7 life-15-00855-f007:**
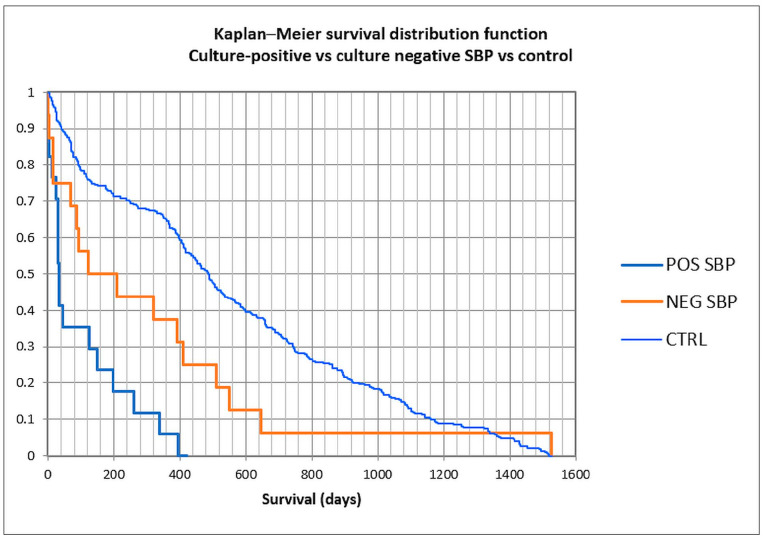
Kaplan–Meier curve in culture-positive, culture-negative spontaneous bacterial peritonitis, and control patients.

**Figure 8 life-15-00855-f008:**
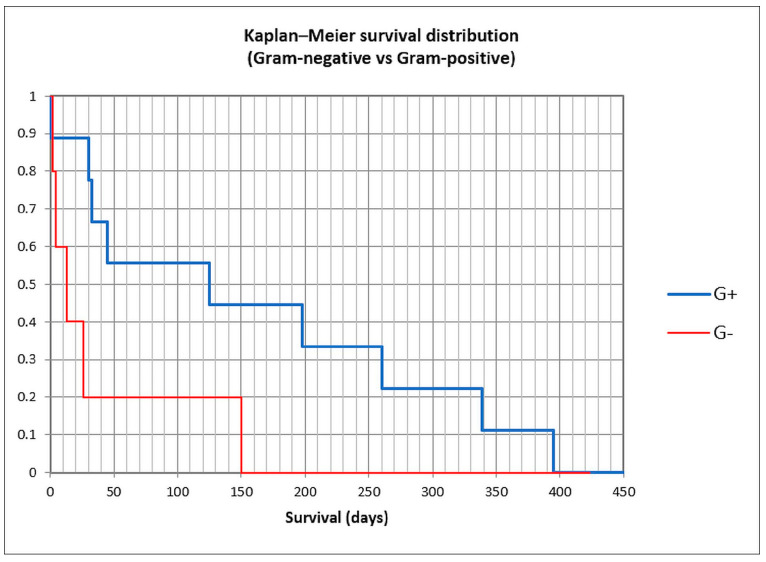
Kaplan–Meier curve in Gram-positive and Gram-negative spontaneous bacterial peritonitis.

**Table 1 life-15-00855-t001:** Characteristics of patients with ascitic fluid infection (N = 51).

Age (Years); Mean ± Standard Deviation (Range)		61.4 ± 11.5 (40–84)
Gender Males/Females (% Males)		33/18 (66)
Cirrhosis etiology (%)		
Alcoholic		39 (76.5)
Hepatitis B		4 (7.8)
Hepatitis B + Alcohol		2 (3.9)
Hepatitis C		4 (7.8)
Hepatitis C + Alcohol		2 (3.9)
Non-nosocomial/Nosocomial infection		26/25 (51/49)
Polymicrobial		11 (21.6)
Recurrent (%)		1 (2)
Clinical diagnosis (%)		
Abdominal pain		25 (49)
Abdominal tenderness		2 (3.9)
Oliguria		6 (11.8)
Dyspnea		6 (11.8)
Fever		0 (0)
Laboratory analyses Median (IQR)		
Hemoglobin (g/dL)		10.40 (9.20–11.71)
Leucocyte count (count/mm^3^)		8100 (5310–13,250)
Neutrophil count (count/mm^3^)		5310 (4395–10,690)
Lymphocyte (count/mm^3^)		1389 (835–1632)
Platelet count (×1000/mm^3^)		138 (99.1–184.6)
Urea (mg/dL)		45 (25–72)
Creatinine (mg/dL)		0.92 (0.70–1.41)
INR		1.69 (1.33–2.05)
Albumin (g/dL)		2.50 (2.05–3.05)
Total bilirubin (mg/dL)		2.73 (1.55–7.22)
ALT (UI/dL)		29 (23–41)
AST (UI/dL)		69 (52–102)
Na (mmol/L)		133 (128–136)
K (mmol/L)		4.5 (3.9–5.1)
Systolic blood pressure mm Hg (Mean ± standard deviation)		116.9 ± 19.2
Pulse (Mean ± standard deviation)		91 ± 18
Child class (%)	A	2 (3.9)
	B	22 (43.1)
	C	27 (52.9)
Child/MELD score (Mean ± standard deviation)		
CTP		9.8 ± 1.8
CTP-creatinine		10.6 ± 2.3
MELD-3		37.9 ± 18.6
MELD-Na		37.3 ± 14.2
Complications (%)		
Encephalopathy		9 (18)
Acute kidney injury		17 (33.3)
Comorbidities (%)		
Hepatocellular carcinoma		7 (14)
Portal vein thrombosis		5 (10)
Acute variceal bleeding		6 (12)
Pneumonia		3 (6)
Clostridium difficile colitis		3 (6)
Cardiovascular		3 (6)
Diabetes		5 (10)
Previous chronic kidney disease		1 (2)
Mortality%	SBP/SFP in-hospital/30-day/90-day/1-year	26.7/44.4/46.1/73
	Bacterascites in-hospital/30-day/90-day/1-year	0/0/16.7/50
	Non-nosocomial SBP/SFP (in-hospital)	14.3
	Nosocomial SBP/SFP (in-hospital)	37.5
Treatment-culture-positive SBP (%)		
Cefoperazone + sulbactam		18 (40)
Ciprofloxacin		6 (13)
Moxifloxacin		4 (8.8)
Levofloxacin		4 (8.8)
Meropenem		3 (6.7)
Vancomycin		3 (6.7)
Imipenem + cilastatin		2 (4.4)
Ertapenem		2 (4.4)
Cefazolin		2 (4.4)
Gentamycin		2 (2.2)
Linezolid		1 (2.2)
Tigecyclin		1 (2.2)
Teicoplanin		1 (2.2)
Norfloxacin		1 (2.2)

IQR = Interquartile Range, INR = International Normalized Ratio, ALT = Alanine Aminotransferase, AST = Aspartate Aminotransferase, CTP = Child-Turcotte-Pugh score, MELD = Model for End-stage Liver Diseases, SBP = Spontaneous Bacterial Peritonitis, SFP = Spontaneous Fungal Peritonitis.

**Table 2 life-15-00855-t002:** Etiology for non-nosocomial- and nosocomial-associated infection.

	Non-Nosocomial Infection (%)	Nosocomial Infection (%)
Gram-positive	21 (65.6)	18 (60)
* Staphylococcus aureus *	13 (40.6)	12 (40)
* Streptococcus * spp.	4 (12.5)	2 (6.7)
* Enterococcus * spp.	3 (9.4)	4 (13.3)
* Staphylococcus coagulase-negative *	1 (3.1)	0 (0)
Gram-negative	9 (28.1)	12 (40)
* E. coli *	3 (9.4)	3 (10)
* Klebsiella * spp.	2 (6.2)	3 (10)
* Acinetobacter baumanii *	1 (3.1)	3 (10)
* Pseudomonas aeruginosa *	1 (3.1)	2 (6.7)
* Citrobacter *	1 (3.1)	0 (0)
* Proteus mirabilis *	1 (3.1)	0 (0)
* Enterobacter * spp.	0 (0)	1 (3.3)
Candida	2 (6.2)	0 (0)
TOTAL	32	30

**Table 3 life-15-00855-t003:** Resistance to antibiotics for the main isolated strains.

	Non-Nosocomial Infection						Nosocomial Infection					
* ANTIBIOTIC *	*Staphylococcus aureus*	*Streptococcus* spp.	*Enterococcus* spp.	*Escherichia coli*	*Klebsiella* spp.	*Acinetobacter baumanii*	*Staphylococcus aureus*	*Streptococcus* spp.	*Enterococcus* spp.	*Escherichia coli*	*Klebsiella* spp.	*Acinetobacter baumanii*
Amikacin	37.5	-	-	-	0	0	75	-	-	0	25 *	100
Aztreonam	-	-	-	0	0	-	-	-	-	0	0	-
Ampicillin	0 *	33.3	0	100	100	-	0 *	-	75	100	100	-
Ampicillin + clavulanic	-	0 *	-	0	50	-	-	-	-	0	60	-
Ampicillin sulbactam	-	-	-	0	100	100	-	-	-	0	75	100
Cefazolin	-	-	-	33.3	100	-	-	-	-	0 *	100	100 *
Cefepime	-	0	-	0	0	100	-	0	-	0	100	100
Cefotaxime	-	0	-	0	-	100	-	0	-	-	100	100
Cefoxitin	-	100 *	-	0	0	-	-	-	-	0	100	-
Ceftaroline	50 *	-	-	-	-	-	50 *	-	-	-	-	100 *
Ceftriaxone	-	50 *	-	0	0	-	-	0	-	0	100	100 *
Cefuroxime	-	-	-	0	0	-	-	-	-	0	100	-
Ceftazidime	-	-	-	0	0	100	-	-	-	0	75	100
Ciprofloxacin	23.1	-	0	33.3	0	100	41.7	-	25	0	100	100
Clarithromycin	53.8	0	-	-	-	-	63.6	0	-	-	-	-
Clindamycin	61.5	0	-	-	-	-	58.3	0	-	-	-	-
Colistin	-	-	-	0 *	0	0	-	-	-	0	0	0
Chloramphenicol	30.8	0	-	0	0	-	11.1	0	-	0	0	-
Levofloxacin	0 *	33.3	0	0 *	-	100	50	-	0	0	-	100 *
Ofloxacin	-	66.7	0 *	-	-	-	-	0	0	-	-	-
Moxifloxacin	16.7	0 *	-	-	-	-	10	0	-	-	-	-
Doxycycline	25	100 *	0 *	-	-	0	18.2	100	-	-	-	0
Nitrofurantoin	-	-	-	-	-	-	-	-	33.3	-	-	-
Quinupristin	0 *	-	-	-	-	-	-	-	-	-	-	-
Linezolid	0	0	0	-	-	-	0	0	0	-	-	-
Erythromycin	69.2	0	-	-	-	-	58.3	100	-	-	-	-
Gentamycin	40 *	-	0 *	50	0	0	30	-	50	0	50	100
Ertapenem	-	0 *	-	0	0	-	-	0	-	0	100	100
Imipenem	-	0 *	-	0	0	0	-	-	-	0 *	66.7	100
Meropenem	-	0 *	-	0	0	0	-	0	-	0	100	100 *
Oxacillin	53.8	100 *	100	-	-	-	54.5	-	100	-	-	-
Penicillin	100	50	0	-	-	-	100	50	75	-	-	-
Piperacillin	-	100 *	-	0	-	0	100 *	-	-	0	100 *	100 *
Rifampicin	25	0 *	-	-	-	-	27.3	-	-	-	-	-
TMP + SMTX	33.3	100 *	-	50	0	-	36.4	100	-	-	66.7	66.7
Teicoplanin	0 *	-	100 *	-	-	-	-	-	0	-	-	-
Tetracycline	40	66.7	0 *	0 *	100	100	40	0	25	0	33.3	100 *
Tigecycline	12.5	0 *	0 *	0	0	0	25	-	-	0	0	0 *
Tobramycin	100 *	-	-	0 *	-	100	-	0	-	0 *	100	50
Vancomycin	-	0	0	-	-	-	0 *		0	-	-	-

TMP + SMTX = Trimethoprim with sulphamethoxazole, * = less than 50% of isolated strains were tested. Higher resistance rates are emphasized by darker shadow cells.

**Table 4 life-15-00855-t004:** Risk factors for Gram-positive and Gram-negative SBP. Only factors with *p* < 0.30 are shown.

	Gram-Positive N = 25	Gram-Negative N = 15	Univariate OR/ Cohen’s d (*p*-Value)	Multivariate OR (95%CI, *p*-Value)
Pulse (beats/min)	85 ± 16 (−12.08)	97 ± 24	0.61 (0.068)	0.99 (0.96–1.04, 0.887)
Normal vs. hypotension	24 (96)	11 (73.3)	8.73 (0.036)	8.12 (0.65–100.29, 0.102)
Encephalopathy	28	13.3	2.53 (0.282)	1.59 (0.16–14.28, 0.693)
Alcoholic/viral (%)	23/2 (92/8)	11/4 (73.3/26.7)	4.18 (0.109)	4.29 (0.37–49.97, 0.245)
Neutrophils (×1000/mm^3^)	5.31 (4.3–10.28)	7.7 (4.61–12.36)	0.49 (0.140)	0.99 (0.09–1.01, 0.117)
Lymphocytes (×1000/mm^3^)	1.51 (0.67–1.82)	1.02 (0.55–1.44)	−0.60 (0.074)	1.01 (0.99–1.01, 0.226)
Platelets (×1000/mm^3^)	138 (111–165)	168 (117.6–250)	0.131	0.99 (0.99–1.01, 0.242)
INR	1.45 (1.33–2.10)	1.77 (1.44–2.12)	0.162	0.44 (0.06–3.24, 0.423)
Direct bilirubin	1.9 (0.93–3.62)	1.63 (0.72–4.12)	0.246	0.98 (0.80–1.21, 0.867)

SBP = Spontaneous Bacterial Peritonitis, OR = Odds Ratio, INR = International Normalized Ratio.

**Table 5 life-15-00855-t005:** Risk factors for death in Gram-positive and Gram-negative SBP.

Factor	Univariate Analysis Odds Ratio (95%CI), *p*	Multivariate Analysis Odds Ratio (95%CI, *p*)
Age < 50/50–69/≥70 years	0.9665	
Male gender	4.33 (0.6207–30.2505), 0.139	15.90 (1.022–247.6, 0.048)
Previous antibiotics	0.82 (0.2101–3.2338), 0.782	
Previous paracentesis	1.06 (0.2504–4.4490), 0.941	
Previous PPI	0.67 (0.1620–2.7430), 0.574	
Hypotension	4.33 (0.7995–23.4877), 0.089	4.73 (0.218–102.818, 0.323)
Child C vs. A/B	1.50 (0.3646–6.1717), 0.574	
AKI	7.33 (1.6333–32.9258), 0.009	8.45 (1.00–71.41, 0.049)
HCC	0.11 (0.0060–2.1830), 0.1450	0.144 (0.004–5.716, 0.302)
PVT	0.92 (0.1521–5.5661), 0.928	
Cardiovascular disease	0.29 (0.0139–6.0896), 0.427	
Diabetes	1.67 (0.2410–11.5250), 0.605	
UGIB	1.53 (0.3017–7.7919), 0.606	
* Clostridium difficile * infection	5.40 (0.4398–66.2977), 0.187	0.610 (0.008–46.378, 0.823)
Alcoholic etiology of cirrhosis	1.67 (0.2918–9.5204), 0.566	
Nosocomial vs. non-	3.46 (0.7701–15.5601), 0.105	10.83 (0.69–170.16, 0.091)
Gram-negative vs. positive	6.00 (1.3742–26.1970), 0.017	7.71 (1.216–48.981, 0.030)

AKI = Acute Kidney Injury, HCC = Hepatocellular Carcinoma, UGIB = Upper Gastrointestinal Bleeding, PPI = Proton Pump Inhibitor, PVT = Portal Vein Thrombosis.

## Data Availability

The data presented in this study are available on request from the corresponding author.

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
