# Peer review of "Predominant Gram-Positive Etiology May Be Associated with a Lower Mortality Rate but with Higher Antibiotic Resistance in Spontaneous Bacterial Peritonitis: A 7-Year Study in a Tertiary Center in Romania"

_life, 2025, doi:10.3390/life15060855_

Round 1

Reviewer 1 Report

Comments and Suggestions for Authors

The work by Cazacu et al., “The microbial shift in spontaneous bacterial peritonitis: a 7-year study in a tertiary center in Romania,” addresses a clinically significant issue that includes 749 subjects screened, with 51 culture-positive cases analyzed for the study. Overall, this is a scientifically valuable study with adequate rigor and relevant findings. The retrospective design is appropriate, and the focus on 7 years time span offers the potential to detect and follow meaningful trends. The authors present a detailed microbiological analysis and provide valuable data. However, some limitations should be addressed by the authors to enhance clarity, strengthen the conclusions before publication.

Here are my overall comments for the authors.

Major comments:

  1. Out of 749 SBP, only 51 were analysed, which excludes a large number of subjects. This is a relatively small number over 7-year span for generalizing trends in resistance patterns and mortality. Further, no information is given about the time distribution, how these cases are distributed across years or concentrated, which affects the validity of “trend claims”. Authors need to address these points. Over seven years (n=51) is relatively small, limiting statistical power and generalizability. It would help to show whether this shift occurred gradually over time or was present throughout. A year-by-year breakdown could make the trend more convincing.
  2. Again, to make a bold claim of microbial shift towards GPB, year on year trend or comparison to historical data would be required to justify the claim. Authors should be cautious about generalizing "trends" from a single-center, retrospective study.
  3. Table 3: The antibiotic resistance data are detailed and thorough. Some antibiotics were tested in very small numbers of isolates, why there is a disparity in the total number of isolates among the antibiotics used in the study.
  4. Some of the tables are dense and hard to follow, summarizing key findings in a figure (heatmap/bar graph) might help readers focus on the most clinically relevant patterns.

Minor comments:

  1. Inclusion of bacterascites cases in the resistance analysis is questionable given their distinct clinical/biological trajectory. It may complicate interpretation, since their clinical course can differ from SBP.
  2. Consider integrating the most clinically important resistance findings into the main text and moving exhaustive data to supplementary material.
Comments on the Quality of English Language

No comments.

Author Response

The work by Cazacu et al., “The microbial shift in spontaneous bacterial peritonitis: a 7-year study in a tertiary center in Romania,” addresses a clinically significant issue that includes 749 subjects screened, with 51 culture-positive cases analyzed for the study. Overall, this is a scientifically valuable study with adequate rigor and relevant findings. The retrospective design is appropriate, and the focus on a 7-year time span offers the potential to detect and follow meaningful trends. The authors present a detailed microbiological analysis and provide valuable data. However, some limitations should be addressed by the authors to enhance clarity, strengthen the conclusions before publication.

Here are my overall comments for the authors.

Major comments:

  1. Out of 749 SBP, only 51 were analyzed, which excludes a large number of subjects. This is a relatively small number over 7 years for generalizing trends in resistance patterns and mortality. Further, no information is given about the time distribution, how these cases are distributed across years or concentrated, which affects the validity of “trend claims”. Authors need to address these points. Over seven years (n=51) is relatively small, limiting statistical power and generalizability. It would help to show whether this shift occurred gradually over time or was present throughout. A year-by-year breakdown could make the trend more convincing.

R: Thank you for your comments. During the 7-year study, 749 patients with cirrhosis and ascites were diagnosed; only 51 had confirmed ascitic fluid infection (from which 6 had bacterascites), 28 had culture-negative SBP, and 670 had no SBP and were used as the control group. We used the term “microbial shift” to emphasize the change from predominant Gram-negative to predominant Gram-positive etiology in patients with cirrhosis and ascitic fluid infection; a temporal trend analysis will require a longer timeframe, such as a 20-year study, which is very difficult to perform (lack of detailed data, more differences in susceptibility testing). From 51 cases with ascitic fluid infection, we had 10 cases during 2018 (6 Gram-positive and 3 Gram-negative), 4 cases in 2019 (2 Gram-positive and 1 Gram-negative), 4 cases in 2020 (1 Gram-positive and 2 Gram-negative), 3 cases in 2021 (2 Gram-positive and 1 Gram-negative), 13 cases in 2022 (8 Gram-positive and 4 Gram-negative), 3 cases in 2023 (2 Gram-positive and none Gram-negative), and 14 cases in 2024 (9 Gram-positive and 5 Gram-negative); the rest of the cases were mixed (Gram-positive and Gram-negative) or fungal peritonitis. Most of the published studies have a low number of SBP (mostly below 100), and a temporal trend is difficult to figure out in studies below 10 years. We added Figure 1 for the illustration of the above-mentioned data.

  1. Again, to make a bold claim of microbial shift towards GPB, a year-on-year trend or comparison to historical data would be required to justify the claim. Authors should be cautious about generalizing "trends" from a single-center, retrospective study.

R: Thank you for your observation. We replaced the term “trend” with “change” to emphasize the change from Gram-negative to Gram-positive etiology in patients with cirrhosis and ascitic fluid infection. Because of the lack of current detailed data in Romania and a predominant Gram-negative etiology in SBP in older data, we presumed that the shift to Gram-positive occurs in Romania during the last decades, as in other countries.

  1. Table 3: The antibiotic resistance data are detailed and thorough. Some antibiotics were tested in very small numbers of isolates, why there is a disparity in the total number of isolates among the antibiotics used in the study.

R: The difference in susceptibility testing was related to the presumed etiology of SBP and the particularities of antibiotic panels used for testing each bacterial species. For that reason, we specified in our table the cases when less than 50% of strains were tested.

  1. Some of the tables are dense and hard to follow; summarizing key findings in a figure (heatmap/bar graph) might help readers focus on the most clinically relevant patterns.

R: Thank you for your observation.  We constructed Figures 2,3, and 4 (Heatmaps) for an easier understanding of the obtained findings. We moved the corresponding tables to the Supplementary Tables 1-8 file.

Minor comments:

  1. Inclusion of bacterascites cases in the resistance analysis is questionable given their distinct clinical/biological trajectory. It may complicate interpretation, since their clinical course can differ from SBP.

R: Thank you for your observation.  Bacterascitis is associated with a different prognosis than SBP; we included all patients with culture-positive SBP and bacterascitis ONLY in the susceptibility analysis, whereas ONLY patients with SBP were included in the analysis of Gram-positive SBP prediction and survival analysis.

  1. Consider integrating the most clinically important resistance findings into the main text and moving exhaustive data to supplementary material.

R: Thank you for your observation.  We moved tables 3 and 4 to the Supplementary Tables 1-8 file.

Reviewer 2 Report

Comments and Suggestions for Authors

Evaluation Report

Abstract and Title
The title accurately reflects the content of the study, but it could be made more specific by explicitly referring to the antimicrobial resistance and mortality findings. The abstract is detailed and well written.

Introduction
The introduction provides adequate background and correctly emphasizes the clinical burden of SBP and the rising issue of antimicrobial resistance. While a general aim is present, a clearly formulated hypothesis or specific research question is not distinctly highlighted. It would benefit from a sharper focus on the study gap this work is filling, ideally supported by recent epidemiological references or the need for this study in this region specifically.

Materials and Methods

Here are several methodological concerns that need to be addressed:

  • The selection criteria limit the study to culture-positive SBP, potentially excluding a significant proportion of clinically relevant culture-negative cases. Might this create a bias?
  • Details regarding the timing of sample collection, microbiological quality control, internal or external quality controls, reproducibility measures, and community differentiation are minimal.
  • Multivariate analysis methods lack clarity concerning variable inclusion, potential collinearity, and checks for model assumptions.
  • Polymicrobial infections are underrepresented; however, the handling of such data in analysis remains unclear.

Overall, more transparency and rigor are needed in both microbiological and statistical components to ensure reproducibility and validity.

Results
The results section is well-structured and comprehensive, presenting both microbiological findings and clinical outcomes. Key weaknesses include:

  • Data tables are so lengthy and not reader-friendly. Some could be visualized with charts or stratified by infection type for clarity.
  • The sample size of certain subgroups (e.g., fungal peritonitis) is too small for robust inference but discussed as if conclusions are definitive.
  • There is no adjustment for multiple testing, which may inflate the significance of some associations.
  • The control group’s comparability is not clearly explained—methods for matching or adjusting baseline variables are needed.

The link between specific pathogens and mortality is intriguing, but given the limited statistical power, it should be interpreted more cautiously.

Discussion
The discussion is broad and well-supported by existing literature. However, several limitations are not sufficiently acknowledged:

  • The retrospective design inherently limits causal inference.
  • Conclusions on evolutionary shifts or causality (e.g., GNB as independent mortality predictors) may be overstated.
  • The suggestion that findings support a redefinition of empirical therapy lacks consideration of local guideline variability and broader resistance surveillance data.
  • Lack of genomic analysis (e.g., WGS for AMR genes) is a missed opportunity and should be discussed as a limitation.

More clearer interpretation would improve scientific accuracy and usefulness to clinicians.

Conclusion

The conclusions align with the general findings of the study, emphasizing the prognostic value of early culture-based therapy in SBP. However, the assertion that Gram-negative infection is an independent predictor of mortality should be tempered, considering the limitations in sample size and potential confounders. The call for changes in empirical treatment should be more conservative and better contextualized within multicenter or prospective evidence.

Author Response

Comments and Suggestions for Authors

Abstract and Title

The title accurately reflects the content of the study, but it could be made more specific by explicitly referring to the antimicrobial resistance and mortality findings. The abstract is detailed and well written.

R: Thank you for your observation. We modified the title to reflect the increased antibiotic resistance along with the changes to Gram-positive etiology, and also with the potential impact of the shift to Gram-positive etiology on mortality.

Introduction
The introduction provides adequate background and correctly emphasizes the clinical burden of SBP and the rising issue of antimicrobial resistance. While a general aim is present, a clearly formulated hypothesis or specific research question is not distinctly highlighted. It would benefit from a sharper focus on the study gap this work is filling, ideally supported by recent epidemiological references or the need for this study in this region specifically.

              R: Thank you for your observation. We have revised the Introduction to clearly state our hypothesis and research question by adding Lines 89-91 to the Introduction section.

Materials and Methods

Here are several methodological concerns that need to be addressed:

  • The selection criteria limit the study to culture-positive SBP, potentially excluding a significant proportion of clinically relevant culture-negative cases. Might this create a bias?

R: We align with your point of view. While microbiological resistance analyses focused on culture-positive SBP, we did include culture-negative SBP in the overall mortality and clinical outcome analyses. We have clarified this in the Materials and Methods and Discussion sections to acknowledge the potential for selection bias and how we attempted to mitigate it (see Lines 148–153 and 539–559).

  • Details regarding the timing of sample collection, microbiological quality control, internal or external quality controls, reproducibility measures, and community differentiation are minimal.

R: Thank you for your observation. We add details regarding the above-mentioned data in lines 128-145.

  • Multivariate analysis methods lack clarity concerning variable inclusion, potential collinearity, and checks for model assumptions.

R: Thank you for your observation. Multicollinearity was evaluated using the variance inflation factor (VIF). We checked if all retained variables had VIF < 2, indicating no significant collinearity. Logistic regression assumptions were verified as follows: Linearity of logit (assessed via Box-Tidwell test), outliers/influential points (evaluated using Cook’s distance), and Goodness-of-fit (Hosmer-Lemeshow test; p > 0.05). To account for multiple testing, we applied the Benjamini-Hochberg FDR correction to univariate p-values. Only results with p less than the FDR threshold were significant after adjustment in the multivariate model. We added this information in Subsection 2.4. Statistical analysis.

We modified Table 6 as follows: total leucocyte count was included by mistake and was removed from the table (the tests reveal positive collinearity with neutrophil count and lymphocyte count). We re-run the statistical analysis and correct the data in multivariate analysis.

  • Polymicrobial infections are underrepresented; however, the handling of such data in analysis remains unclear.

R: Thank you for your observation.  Polymicrobial infections with two Gram-positive or two Gram-negative strains were included in the Gram-positive and Gram-negative subgroups, whereas those with mixed strains (one Gram-positive and one Gram-negative) were not included in the analysis between Gram-positive and Gram-negative SBP.

Overall, more transparency and rigor are needed in both microbiological and statistical components to ensure reproducibility and validity.

Results
The results section is well-structured and comprehensive, presenting both microbiological findings and clinical outcomes. Key weaknesses include:

  • Data tables are so lengthy and not reader-friendly. Some could be visualized with charts or stratified by infection type for clarity.

R: Thank you for your observation.  We constructed Figure 3-5 (Heatmaps) to better illustrate the findings. We moved the corresponding tables to the Supplementary Tables 1-8 file.

  • The sample size of certain subgroups (e.g., fungal peritonitis) is too small for robust inference but discussed as if conclusions are definitive.

R: We acknowledge that limitation. However, fungal peritonitis was briefly mentioned in the Results section; in the Discussions section, we analyzed the percentage of the total patients with ascitic fluid infection, the potential risk factors, treatment particularities, and long-term evolution. No statistical analysis was possible because of the small size of this subgroup of patients.

  • There is no adjustment for multiple testing, which may inflate the significance of some associations.

R: Thank you for your observation. To account for multiple testing, we applied the Benjamini-Hochberg FDR correction to univariate p-values. Only results with p less than the FDR threshold (0.30) were significant after adjustment in the multivariate model.

  • The control group’s comparability is not clearly explained—methods for matching or adjusting baseline variables are needed.

R: Thank you for your observation.  In univariate and multivariate analyses, the whole group of patients with cirrhosis and non-infected ascites (670 patients) was used as the control group. Because all the demographic and clinical parameters were introduced in the univariate and multivariate analyses, no matching or adjustment was made.

The link between specific pathogens and mortality is intriguing, but given the limited statistical power, it should be interpreted more cautiously.

              R: Thank you for your observation. We added Lines 531-532 and 539-559 to emphasize that limitation.

Discussion
The discussion is broad and well-supported by existing literature. However, several limitations are not sufficiently acknowledged:

  • The retrospective design inherently limits causal inference.
  • Conclusions on evolutionary shifts or causality (e.g., GNB as independent mortality predictors) may be overstated.
  • The suggestion that findings support a redefinition of empirical therapy lacks consideration of local guideline variability and broader resistance surveillance data.
  • Lack of genomic analysis (e.g., WGS for AMR genes) is a missed opportunity and should be discussed as a limitation.

More clearer interpretation would improve scientific accuracy and usefulness to clinicians.

R: Thank you for your comments and suggestions. We added these limitations to our study (Lines 539-559).

Conclusion
The conclusions align with the general findings of the study, emphasizing the prognostic value of early culture-based therapy in SBP. However, the assertion that Gram-negative infection is an independent predictor of mortality should be tempered, considering the limitations in sample size and potential confounders. The call for changes in empirical treatment should be more conservative and better contextualized within multicenter or prospective evidence.

R: Thank you for your observation; the small patient group and retrospective design may significantly impact the generalizability of our findings regarding the potential role of Gram-negative versus Gram-positive bacteria in prognosis assessment in patients with SBP. We added/modified Lines 574-580 in the Conclusions section to better integrate the obtained findings in the current data milieu.

Reviewer 3 Report

Comments and Suggestions for Authors

Fairly complete description of clinical of SBP over 7 years at a single institution over 7 years. There are several features of this review which could be improved:

  1. the title suggests a shift over a 7 year period, but no data provided to support a change in microbiology within that time span.  The title should reflect simply the relative increase in GBB vs GNB compared to the literature
  2. risk factor analysis should include the cirrhosis etiology, not just alcoholism, but viral etiology (all types)
  3. far too much emphasis on antimicrobial susceptibility testing (AST), both in text and tables.  Tables 3, 4, and 5 not useful, especially due to the small numbers of isolates outnumbered by the number of drugs tested with highly variable denominators.  Basic rule is not to present % S, I, R unless you have at least 20 isolates of a single genus/species.
  4. No microbiological outcomes presented. No follow-up cultures??  
  5. Use of the term "efficient" or efficiency" to suggest clinical efficacy is not supported by your outcomes data.  All you have is in vitro AST.
  6. Conclusions not supported by data regarding antibiotics.  Particularly the suggestion that colistin could be useful.  Colistin in countries other than China, is CMS, a prodrug, that needs to be converted to active drug by the kidneys. It is also nephrotoxic.  Given that many patients have AKI or hepatorenal syndrome with their cirrhosis, using a nephrotoxic drug, even polymyxin B, is not the best choice.
  7. although your numbers are small, the relatively high rate of S. aureus and the absence of S. pneumoniae does represent a change in etiologic spectrum from past literature
  8. minor point: all antibiotics are generic names and should not be capitalized

Author Response

Response to Reviewer 3

Yes

Can be improved

Must be improved

Not applicable

Does the introduction provide sufficient background and include all relevant references?

( )

(x)

( )

( )

Is the research design appropriate?

( )

(x)

( )

( )

Are the methods adequately described?

( )

(x)

( )

( )

Are the results clearly presented?

( )

( )

(x)

( )

Are the conclusions supported by the results?

( )

( )

(x)

( )

Comments and Suggestions for Authors

Fairly complete description of clinical of SBP over 7 years at a single institution over 7 years. There are several features of this review which could be improved:

  1. The title suggests a shift over a 7 year period, but no data provided to support a change in microbiology within that time span.  The title should reflect simply the relative increase in GBB vs GNB compared to the literature.

R: Thank you for your observation. We modified the title to reflect the increased antibiotic resistance along with the changes to Gram-positive etiology, and also with the potential impact of the shift to Gram-positive etiology on mortality.

  1. Risk factor analysis should include the cirrhosis etiology, not just alcoholism, but viral etiology (all types).

R: We analyzed the etiology as a risk factor (alcohol, viral B or C, and mixed viral+alcohol). In univariate analysis, mixed etiology had an OR close to viral etiology, so further statistical analysis has included mixed etiology together with viral etiology. No separate analysis was possible for B or C etiology because of the small number of positive cases.

  1. far too much emphasis on antimicrobial susceptibility testing (AST), both in text and tables.  Tables 3, 4, and 5 not useful, especially due to the small numbers of isolates outnumbered by the number of drugs tested with highly variable denominators.  Basic rule is not to present % S, I, R unless you have at least 20 isolates of a single genus/species.

R: We agreed to that limitation. However, in most studies regarding SBP, the sample size is between 50 and 100 cases (References 18,19,25,27,28,33,34,36,39,40,55,57), with only a few studies showing more than 20 isolated strains for one bacterial type. We consider that a detailed analysis of the most frequently isolated species represents an important guide for antibiotic choice, especially given the geographic susceptibility variability and paucity of the data for some countries. A table presenting the percentage of resistance (with the limitation for testing mentioned below the table) may aid in the management of SBP, even though no isolated bacterial type had 20 or more isolated strains.

  1. No microbiological outcomes presented. No follow-up cultures??  

R: Thank you for your question. The second paracentesis was available in only 8 cases, the main purpose being the reevaluation of ascitic neutrophil count. All bacteriological samples were negative.

  1. Use of the term "efficient" or efficiency" to suggest clinical efficacy is not supported by your outcomes data.  All you have is in vitro AST.

R: We acknowledge that limitation. We changed the terms ”efficient” or ”efficiency” with susceptibility.

  1. Conclusions not supported by data regarding antibiotics.  Particularly the suggestion that colistin could be useful.  Colistin in countries other than China, is CMS, a prodrug, that needs to be converted to active drug by the kidneys. It is also nephrotoxic.  Given that many patients have AKI or hepatorenal syndrome with their cirrhosis, using a nephrotoxic drug, even polymyxin B, is not the best choice.

R: We acknowledge the limitations of colistin use in patients with cirrhosis and AKI or hepato-renal syndrome. We modified the conclusions to emphasize this limitation (Lines 569-571).

  1. Although your numbers are small, the relatively high rate of S. aureus and the absence of S. pneumoniae does represent a change in etiologic spectrum from past literature.

R: Thank you for this observation. The etiological change in patients with SBP is indeed interesting and may alter current local guidelines for SBP management.

  1. Minor point: all antibiotics are generic names and should not be capitalized

R: We changed all antibiotic names according to your observation.

Round 2

Reviewer 1 Report

Comments and Suggestions for Authors

The authors have addressed all my questions and concerns regarding the manuscript's scientific soundness and clarity. The manuscript can be accepted in its current form.